# Anti-Biofilm Activity of Cannabidiol against *Candida albicans*

**DOI:** 10.3390/microorganisms9020441

**Published:** 2021-02-20

**Authors:** Mark Feldman, Ronit Vogt Sionov, Raphael Mechoulam, Doron Steinberg

**Affiliations:** 1Biofilm Research Laboratory, Faculty of Dental Medicine, Institute of Dental Sciences, The Hebrew University of Jerusalem, Jerusalem 9112002, Israel; ronitsionov@gmail.com (R.V.S.); dorons@ekmd.huji.ac.il (D.S.); 2Medicinal Chemistry Laboratory, The Institute for Drug Research, School of Pharmacy, Faculty of Medicine, The Hebrew University of Jerusalem, Jerusalem 9112002, Israel; raphaelm@ekmd.huji.ac.il

**Keywords:** *C. albicans*, CBD, biofilm, virulence, cell stress, gene expression

## Abstract

*Candida albicans* is a common fungal pathogen in humans. Biofilm formation is an important virulence factor of *C. albicans* infections. We investigated the ability of the plant-derived cannabidiol (CBD) to inhibit the formation and removal of fungal biofilms. Further, we evaluated its mode of action. Our findings demonstrate that CBD exerts pronounced time-dependent inhibitory effects on biofilm formation as well as disruption of mature biofilm at a concentration range below minimal inhibitory and fungicidal concentrations. CBD acts at several levels. It modifies the architecture of fungal biofilm by reducing its thickness and exopolysaccharide (EPS) production accompanied by downregulation of genes involved in EPS synthesis. It alters the fungal morphology that correlated with upregulation of yeast-associated genes and downregulation of hyphae-specific genes. Importantly, it represses the expression of *C. albicans* virulence-associated genes. In addition, CBD increases ROS production, reduces the intracellular ATP levels, induces mitochondrial membrane hyperpolarization, modifies the cell wall, and increases the plasma membrane permeability. In conclusion, we propose that CBD exerts its activity towards *C. albicans* biofilm through a multi-target mode of action, which differs from common antimycotic agents, and thus can be explored for further development as an alternative treatment against fungal infections.

## 1. Introduction

The fungus *Candida albicans* is a commensal microorganism found in various sites of the human body such as the oral cavity, vagina, and the lungs. It is the most common fungal pathogen in humans, often associated with serious invasive mucosal and systemic infections. One of the most important virulence properties of *C. albicans* is attributed to biofilm formation on either biotic or abiotic surfaces [1,2,3,4]. Formation of biofilm is a complex process that involves attachment to the surface, colonization, and maturation. Biofilms consist of highly structured multilayer network of various cell types: yeast-form cells, pseudohyphal cells, and hyphal cells enwrapped in a protective extracellular matrix [5,6,7,8]. The matrix components include glycoproteins, carbohydrates, lipids, nucleic acids and exopolysaccharides (EPS) [9].

The resistance rate of *C. albicans* biofilms to the majority of antifungal drugs is much higher as compared to the planktonic form of fungi, making biofilm-associated infections difficult to treat. In addition, many of the currently used antifungal drugs have the disadvantages of being highly toxic, the possibility of interacting with other drugs, and the need for intravenous administration [10,11]. Therefore, there is an urgent need for alternative compounds capable to treat fungal biofilms. 

Cannabidiol (CBD) is a non-psychoactive phytocannabinoid produced by the *Cannabis sativa* plant that has been shown to be well-tolerated in human [12,13]. CBD has strong anti-inflammatory properties and is effective in the treatment of various diseases including inflammatory and neurodegenerative diseases, autoimmune diseases, cardiovascular diseases, and cancer [14,15]. Several studies demonstrated an antimicrobial activity of cannabinoids. *C. sativa* extracts showed microbicide activity on both gram-positive and on gram-negative bacteria as well as on some fungi, such as *Aspergillus niger* [16,17]. It has been suggested a potential role for CBD as an antimicrobial agent [18,19,20,21]. CBD was shown to have potent anti-bacterial activity against methicillin-resistant *Staphylococcus aureus* (MRSA) and *Streptococcus* isolates with MICs in the range of 1–5 μg/mL [19,20,22]. Its related cannabigerol (CBG) interfered with quorum sensing (QS) in *Vibrio harveyi* with no detectable minimal inhibitory concentration (MIC) [23]. Pharmacological components of the endocannabinoid system include endocannabinoids (EC), enzymes that synthesize and degrade the EC, and cannabinoid receptors (CB). Our previous findings demonstrated anti-biofilm activity of the ECs anandamide (AEA) and arachidonoyl serine (AraS) against three MRSA strains while exhibiting poor bactericidal activity against planktonic MRSA [24]. Furthermore, ECs exhibited pronounced synergistic effects in combination with either antibiotics or poly-L-lysine on MRSA [25] and *Streptococcus mutans* [26], respectively. Recently, we reported an antifungal activity of AEA and AraS against *C. albicans* where they prevent the adhesion of the fungal hyphae to epithelial cells, and inhibit yeast-hyphae transition and hyphal growth without affecting *C. albicans* viability [27]. The aim of the present study was to determine the potential anti-biofilm activity of CBD and to investigate its mode of action against the fungal pathogen *Candida albicans*.



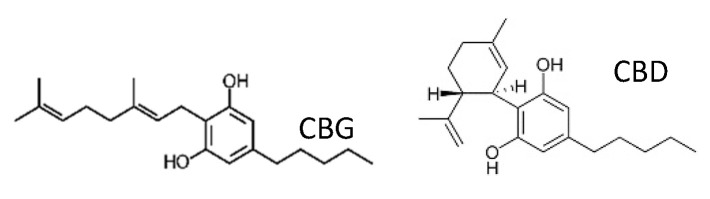



## 2. Materials and Methods

### 2.1. CBD

CBD powder (Figure 1) (purity 99.4%, CAS Number: 13956-29-1) was purchased from NC labs (Czech Republic) and prepared as 10 mg/mL stock solution in ethanol.

### 2.2. Fungal Strains and Growth Conditions

*C. albicans* SC5314 and *C. albicans* SC5314 carrying the green fluorescence protein (GFP) reporter gene (*C. albicans*–GFP) [28] kindly provided by J. Berman (Tel Aviv University, Israel), were grown for 24–48 h at 30 °C on Potato Dextrose Agar (PDA) (Neogen, Lansing, MI) plates. The yeast cells were resuspended at an OD_600_ = 0.05 in RPMI medium (Sigma-Aldrich, St. Louis, MO, USA) and used for biofilm assays by incubation at 37 °C. Control samples were treated with corresponding ethanol concentrations. Blank samples were CBD in RPMI in the absence of fungi.

### 2.3. C. albicans Biofilm Formation

Fungal biofilms were prepared on flat-bottomed 96-well polystyrene tissue culture-grade microtiter plates. *C. albicans* were incubated in RPMI in the presence of increasing concentrations of CBD (1.56 to 100 µg/mL) for 24, 48, and 72 h at 37 °C. After biofilm formation, the wells were washed twice with PBS and used for the different assays described below. To investigate the effect of CBD on preformed biofilms, *C. albicans* biofilms were allowed to mature for 24 h at 37 °C. Thereafter, the wells were washed twice with PBS and fresh RPMI medium containing CBD (3.12–100 µg/mL) was added and the plate was incubated for another 24 h at 37 °C. The assay was performed in triplicates.

### 2.4. Morphological Studies

The biofilms were formed by GFP-expressing *C. albicans* without or with CBD at 25 µg/mL and 100 µg/mL in 24-well polystyrene microtiter plates. Thereafter, the morphology of the washed biofilms was visualized using NIKON confocal microscope (Nikon Inc. Melville, NY, USA) with 488 nm green laser excitation and collecting the data using the 535 nm filter. At least three random fields were captured for each sample. Three independent experiments were performed, and one set of representative data is shown.

### 2.5. Determination of Minimal Inhibitory Concentration (MIC)

The MIC values of CBD against *C. albicans* were determined using the twofold serial microdilution method based on the CLSI protocol [24]. The fungi were incubated with various concentrations of CBD (3.25–400 µg/mL) in RPMI in 96-well plates for 24 h at 37 °C. The MIC was determined as the lowest concentration of CBD showing no turbidity after 24 h, where turbidity was interpreted as visible fungal growth. In addition, the metabolic activity of the planktonic fungi was determined quantitatively using a standard 3-(4,5-dimethyl-2-thiazolyl)-2,5-diphenyl-2H-tetrazolium bromide (MTT) reduction assay (Calbiochem, Burlington, MA, USA) in an Infinite M200 PRO plate reader (Tecan Group Ltd., Männedorf, Switzerland). The assay was performed in triplicates.

The metabolic activity of biofilms was measured by adding 50 μL of 0.5 mg/mL MTT in PBS on the biofilms followed by a 2 h incubation at 37 °C [24]. The amount of intracellular tetrazolium salts was quantified spectrophotometrically by measuring the absorbance at 570 nm.

### 2.6. Determination of Minimal Fungicidal Concentration (MFC)

A total of 10 µL of overnight grown *C. albicans* inoculum exposed to CBD in 96-well plate as described above was plated on PDA and incubated at 37 °C for 3–4 days. Untreated fungi served as positive control. No colonies appearance on agar plates indicates fungicidal effect of CBD. The assay was performed in triplicates.

### 2.7. Determination of Chitin Content in Biofilms

*C. albicans* biofilms were formed in 8-well chamber slide (ibidi GmbH, Gräfelfing, Germany) in the absence or presence of CBD (6.25, 12.5, 25, and 50 μg/mL), washed and stained with 1 μg/mL of Calcofluor White M2R (CFW) (Sigma-Aldrich) for 5 min in the dark. CFW is a fluorophore which emits blue fluorescence after binding to chitin in the fungal cell wall [29]. Total chitin content was measured using the Infinite M20 PRO plate reader (Tecan) (excitation 365 nm, emission 435 nm). Data are presented as percentage of untreated control. In parallel, the CFW-stained biofilms were visualized by NIKON confocal microscope using the DAPI filter set. At least three random fields were observed. Three independent experiments were performed and one set of representative data is shown.

### 2.8. Confocal Laser Scanning Microscopy (CLSM) of Biofilms

*C. albicans*–GFP biofilms were allowed to form in 24-well polystyrene microtiter plates in the absence or presence of 50 μg/mL CBD in RPMI for 24 h at 37 °C. The washed biofilms were incubated for 45 min in PBS containing 25 μg/mL concanavalin A (ConA)-Alexa Fluor 647 conjugate (excitation wavelength 650 nm and emission at 668 nm) (Invitrogen, Carlsbad, CA, USA). ConA binds to glucose and mannose residues of cell wall exopolysaccharides (EPS) [30]. Stained biofilms (green color for viable fungal cells and blue color for EPS) were observed with a NIKON confocal microscope. Three-dimensional images of the formed biofilms and EPS distribution were constructed using the NIS-Element AR software. At least three random fields were captured and analyzed. The amount of total EPS production and viable cells in each sample was calculated according to the fluorescence intensity using Image J (version 3.91; Java image processing program; Rasband, W.S., ImageJ, U. S. National Institutes of Health, Bethesda, MD, USA). The data are presented as total EPS production or total viable cells in each layer of biofilm (5 µm). The percentage of total EPS production or total viable cells in biofilms treated with CBD is presented as area under the curve (AUC) and compared to untreated control. Three independent experiments were performed and one set of representative data is shown.

### 2.9. Assessment of Mitochondrial Function

The mitochondrial membrane potential (MMP) of *C. albicans* was determined in a microtiter well-based assay using the cationic dye 3,3′-diethyloxacarbocyanine iodide (DiOC2(3); Molecular Probes, Eugene, OR, USA). It was demonstrated that this dye localizes itself in the yeast mitochondrial membrane, and thus could be utilized for MMP measurement [31]. Briefly, washed biofilms were resuspended in PBS to a final concentration of OD_600_ = 3 and incubated with 3 μM DiOC2(3) at room temperature for 15 min. A total of 10 μL aliquots of the labeled fungi were diluted in 40 μL PBS containing various concentrations of CBD and read immediately in an Infinite M200 PRO plate reader (Tecan) using excitation at 488 nm and emission at 690 nm. The difference in red fluorescence at 690 nm between CBD-treated and untreated control cells was calculated as percentage of the untreated control.

For mitochondria visualization, biofilms were formed in 8-well chamber slide (ibidi GmbH) and stained with 250 nM of MitoTracker Red CMXRos (Invitrogen , Waltham, MA, USA) a red fluorescent dye that stains mitochondria in live cells and its accumulation is dependent upon membrane potential. The images were captured by NIKON confocal microscope using the 561 nm yellow–green laser excitation and collection of the data using the 635 nm filter. At least three random fields were observed. Three independent experiments were performed and one set of representative data is shown.

### 2.10. Effect of CBD on Intracellular ATP Levels in Biofilm

The biofilms formed by *C. albicans* in the absence or presence of CBD (6.25, 12.5, and 25 μg/mL) as described above were washed with PBS and resuspended in PBS to a final density of OD_600_ = 0.1. A total of 100 μL cell suspension was mixed with the same volume of BacTiter-Glo^TM^ reagent (Promega, Madison, WI, USA) and incubated for 5 min at room temperature prior to reading the luminescence in the Infinite M200 PRO plate reader (Tecan). Data are presented as percentage of the untreated control. The assay was performed in triplicates.

### 2.11. Effect of CBD on ROS Accumulation

Untreated and CBD-treated *C. albicans* biofilms formed in 96-well black microtiter plates grown for 24 h were washed with PBS and exposed to 10 µM of 2′,7′-dichlorofluorescein diacetate (DCFHDA) (Sigma-Aldrich) [32]. DCFHDA passively diffuses through the cell membrane into the cell where it is deacetylated by esterases to form the non-fluorescent 2,7-dichlorofluorescein (DCFH). DCFH reacts with ROS to form the fluorescent product 2,7-dichlorofluorescein (DCF) [33]. Following incubation for 1 h at 37 °C, the biofilms were washed with PBS and the fluorescence intensities (FIs) of the biofilms were measured in an Infinite M200 PRO plate reader (Tecan) (excitation, 485 nm, emission, 535 nm). The FI values were normalized to the amount of metabolically active cells in biofilms assessed by the MTT assay described above. Data are presented as percentage of the untreated control. Fluorescence images were taken using a NIKON confocal microscope with 488 nm green laser excitation and collection of the data using the 535 nm filter. At least three random fields were observed. Three independent experiments were performed and one set of representative data is shown.

### 2.12. Evaluation of Membrane Permeability of Biofilm Cells

The membrane-disrupting activities of CBD on 24 h-old *C. albicans* biofilm were determined using the membrane-impermeable dye propidium iodide (PI) (Sigma-Aldrich). The biofilms were washed with PBS and incubated with 10 μg/mL PI in the dark for 20 min. PI fluorescence was measured in an Infinite M200 PRO plate reader (Tecan) (excitation 535 nm, emission 617 nm). The ratios between dead cells and metabolically active cells in biofilms was calculated as PI fluorescence values divided on values obtained from MTT assay (PI/MTT ratio). The data were presented as a fold increase in PI/MTT ratio in treated biofilms compared to PI/MTT ratio in untreated biofilms equaling 1. In parallel, the PI-stained biofilms were observed by NIKON confocal microscope using the 561 nm yellow-green laser excitation and collecting the data using the 635 nm filter. At least three random fields were observed. Three independent experiments were performed, and one set of representative data is shown.

### 2.13. Quantitative Real Time RT-PCR Analysis of C. albicans Specific Genes

Total RNA was extracted and purified from untreated and CBD (25 µg/mL)-treated fungal biofilms using Tri-Reagent (Sigma-Aldrich) as described [7]. The purified RNA was reverse transcribed to cDNA using the qScript cDNA synthesis kit (Quantabio, Beverly, MA, USA). Real-time qPCR was performed in a CFX96 BioRad Connect Real-Time PCR apparatus using Power Sybr Green Master Mix (Applied Biosystems, Waltham, MA, USA) on 50 ng cDNA and 300 nM of the respective primer sets (Table 3). The relative expression levels of the target genes were analyzed using a Bio-Rad CFX Maestro software (Quantabio, Beverly, MA, USA). Primers for the tested genes were taken from the literature or designed using Primer-BLAST software (https://www.ncbi.nlm.nih.gov/tools/primer-blast/; access on 18 February 2021). For each set of primers, a standard amplification curve (critical threshold cycle vs. exponential of concentration) was plotted, and only those with a slope ≈ −3 were considered reliable. The PCR conditions consisted of a denaturation step at 95 °C for 10 min, followed by 40 cycles of amplification (95 °C for 15 s, 60 °C for 60 s). The expression of *ACT1* gene was used as internal standard. Gene expression is given in relative values, setting the expression level of the untreated control to 1 for each gene. The assay was performed in triplicates.

### 2.14. Statistical Analysis

Means of three independent experiments ± standard errors of the means (SEM)**** were calculated. The statistical analysis was performed using Student’s *t*-test with a significance level of *p* < 0.05 as compared to controls.

## 3. Results

### 3.1. CBD Inhibits Biofilm Formation of Candida Albicans

We initially studied the effect of CBD on biofilm formation. For this purpose, *C. albicans* was exposed to various concentrations of CBD for 24–72 h, and the metabolic activity of the biofilms was analyzed using the MTT assay. We observed a time- and dose-dependent inhibition of fungal biofilm formation by CBD (Figure 1A). After a 24 h of incubation, 12.5 μg/mL CBD inhibited biofilm formation by 37% in comparison to untreated control, while after 72 h, 1.56 μg/mL CBD was sufficient to achieve a similar impact (31% inhibition) (Figure 1A). CBD at a dose of 6.25 μg/mL did not reduce the biofilm mass after 24 h, while an inhibition of biofilm mass by 28% and 39% was observed after 48 h and 72 h, respectively (Figure 1A). Similarly, increasing the CBD dose to 25 μg/mL caused a pronounced time-dependent inhibitory effect. The amount of metabolically active cells in 24 h-, 48 h- and 72 h-biofilms decreased at this dose by 48%, 64%, and 87%, respectively (Figure 1A). CBD exhibited minimal biofilm inhibitory concentration (MBIC) of 90% (MBIC_90_) at 100 μg/mL based on the finding that *C. albicans* biofilm was almost totally inhibited at this concentration at all tested time points (Figure 1A). No minimal inhibitory concentration (MIC) and no minimal fungicidal concentration (MFC) were detected at the tested doses of CBD towards planktonic *C. albicans* [34].

### 3.2. CBD Disrupts Mature Biofilm

Next, we wanted to study the effect of CBD on pre-formed biofilms. Biofilms were allowed to form by an overnight incubation, then washed and exposed to increasing concentrations of CBD. Pre-formed fungal biofilm was dose-dependently affected by CBD. Already at the low dose of 1.56 μg/mL, mature biofilm was decreased by 28% as compared to untreated control (Figure 1B). Increasing the CBD dose to 3.12 μg/mL further enhanced its anti-biofilm activity by disrupting biofilm by 44% (Figure 1B). Finally, the metabolic activity in mature biofilm dropped to 32–46% when treated with higher concentrations (6.25–100 μg/mL) of CBD (Figure 1B).

### 3.3. CBD Alters Biofilm Morphology and Fungal Cell Structure

Control biofilms which were 24 h old uniformly consisted of true hyphae firmly embraced in matrix (Figure 2A). In contrast, biofilm treated with 25 μg/mL CBD appeared heterogeneous with clearly visible numerous cell aggregates of *C. albicans*, while the presence of true hyphae was reduced in comparison to control (Figure 2B). Biofilm exposed to 100 μg/mL CBD contained sporadically spread single cells (Figure 2C).

Higher magnification analysis of cell morphology and structure of calcofluor white (CFW)-stained biofilms revealed a dose-dependent reduction in chitin fluorescence staining intensity with increasing concentrations of CBD [Figure 3(Ab’–Ae’)]. The CBD-treated hyphae appeared with thinner cell wall and septa [Figure 3(Ab’–Ae’)] than the control fungi (Figure 3(Aa’)). A similar pattern of CBD activity was observed after calculating the total chitin synthesis (Figure 4B). At sub-MBIC_90_ of 12.5 and 25 μg/mL, CBD reduced chitin content by 37% and 53%, respectively, while CBD at a dose of 50 μg/mL was able to diminish total chitin composition in biofilm by 90% as compared to the untreated control (Figure 3B). Along with the reduction in chitin content, the morphology of the fungal cell was notably altered by CBD. While control fungi [Figure 3(Aa,a’)] represented uniform true hyphae with true septa (red arrows), CBD-treated fungi [Figure 3(Ab–e,b’–e’)] appeared as pseudohyphae characterized by long elliptic branched forms with constrictions at the septal junctions (white arrows). In addition, true hyphae in the untreated control displayed parallel-sided walls [Figure 3(Aa,Aa’)], whereas the width of each segment of the pseudohyphae in CBD-treated biofilms was uneven, being wider at the center than at the septum [Figure 3(Ab–Ae,Ab’–Ae’)].

The strong reduction in biofilm mass and EPS content after CBD treatment was further proved using confocal laser scanning microscopy (CLSM) (Figure 4D–F vs. Figure 4A–C). CBD reduced the biofilm depth 2.5-fold in comparison to control (40 vs. 100 µm) (Figure 4D–F vs. Figure 4A–C). Image J software analysis showed notable decrease in amount of viable fungal cells (by 85%) as well as EPS production (by 75%) in fungal biofilm treated with 50 μg/mL of CBD in comparison to untreated control (Figure 4G–I).

### 3.4. CBD Alters Mitochondrial Activity and Elevates Intracellular Reactive Oxygen Species (ROS)

Exposure of fungal biofilm to CBD caused significant dose-dependent decrease in the intracellular ATP level. Already at the lowest tested dose of CBD (6.25 μg/mL), the intracellular ATP concentration dropped by 80% in comparison to untreated control (Figure 5C). Increasing the CBD concentration to 25 μg/mL led to an even more pronounced inhibition of ATP synthesis by 90% as compared to control (Figure 5C). Along with this observation, we detected a notable dose-dependent increase of mitochondrial membrane potential (MMP) and intracellular ROS production upon CBD treatment. CBD in the dose range of 6.25–25 μg/mL enhanced MMP and ROS level 1.5–1.8-fold and 1.8–2.8-fold, respectively, in comparison to the untreated control (Figure 5A,B). The elevated MMP and ROS levels were further confirmed by CLSM imaging. To monitor the effect of CBD on MMP, the biofilms were exposed to MitoTracker red CMXRos that stains mitochondria in live cells and its accumulation is dependent upon membrane potential. Mitochondria red staining is relatively weak in control biofilms [Figure 5(Aa,a’)]. However, CBD at 12.5 μg/mL [Figure 5(Ab,b’)] or 25 μg/mL (Figure 5(Ac,c’)) led to strong mitochondrial staining which suggests that CBD causes a raise in MMP. Biofilms that have been stained with the ROS indicator 2′,7′-dichlorofluorescein diacetate (DCFHDA) demonstrated an increase in green fluorescence intensity upon treatment with 6.25 μg/mL and 25 μg/mL CBD [Figure 5(Bb) and Figure 5(Bc)] in comparison to the untreated control that showed low green fluorescence intensity [Figure 5(Ba)], indicating that CBD induces intracellular ROS production in *C. albicans* biofilms.

### 3.5. CBD Causes Membrane Damage to the Fungi in Biofilm

Next, we investigated whether CBD affects the membrane permeability of the fungi. For this purpose, untreated and CBD-treated biofilms were exposed to propidium iodide (PI) that only penetrates the cells if the cell membrane is damaged. PI binds to nucleic acids and emits red fluorescence. No significant red fluorescence was detected in control biofilms (Figure 6A), while a dose-dependent elevation of red fluorescence signal was observed in CBD-treated biofilms (Figure 6B–E). The ratios of dead cells vs. metabolically active cells in the biofilm increased with raising concentrations of CBD (Figure 6F). At MBIC_90_ = 100 μg/mL, CBD induced severe disruption of membranes in biofilm cells which was manifested in massive leakage of PI-stained nuclear compounds from the damaged cells (Figure 6E).

### 3.6. CBD Differentially Affects Gene Expression in Fungal Biofilm

It was important to study whether the inhibitory effect of CBD on biofilm formation is related to changes in gene expression. To this end, the RNA was extracted from untreated and CBD-treated biofilms and analyzed by real-time qPCR. Data obtained from real-time qPCR revealed a variety of *C. albicans* genes whose expression was affected differently upon CBD treatment. CBD at ¼ MBIC_90_ downregulated the expression of genes involved in biofilm maintenance, development, and maturation of factors associated with EPS synthesis such as: *ADH5*, *FKS1* and *BIG1* (1.4–3.75-fold); filamentation: *ECE1*, *HWP1*, *EED1*, *ECE1*, *RAS1*, and *EFG1* (1.46–6.2-fold); adhesion: *ALS3* (1.72-fold) as compared to control (Table 1). Transcripts of genes involved in the regulation of cell wall synthesis and cell separation (*CHS1*, *CHT1*, *CHT2*, *CHT3*) were reduced by CBD to various degrees, ranging from 1.75-fold to 8-fold in comparison to control (Table 1). Genes responsible for the maintenance of the integrity, rigidity, and fluidity of the plasma membrane (*ERG11*, *ERG20*) were downregulated 3.3-fold by CBD when compared to control (Table 1). Transcription of hyphae-specific cell wall proteins were also notably reduced by CBD. Expression of *RBT1*, *RBT4*, and *RBT5* was downregulated 3–4 fold, and expression of *PRA1* was downregulated by more than 9-fold when compared to control (Table 1). Expression of genes associated with antioxidant defense (*SOD1*, *SOD2*, *SOD4*, *TRR1*) was decreased by CBD to different extents, ranging from 2.4-fold to 8-fold when compared to control (Table 1). In addition, *VCX1* encoding for H^+^/Ca^2+^ exchanger was downregulated 2.8-fold as compared to control (Table 1). Importantly, genes involved in *C. albicans* host invasion (*LIP2*, *LIP4*, *LIP5*, *PLB1*, *PLB2*) were downregulated 1.5–3-fold as compared to control (Table 1).

There were also several genes upregulated in the biofilm following exposure to CBD. The expression level of the heat shock proteins HSP70, HSP90, and HSP104 was increased about 2.5-fold by CBD when compared to control (Table 2). Notably, CBD caused a dramatic elevation of the transcript of yeast wall protein (YWP1) by more than 40-fold when compared to control (Table 2). Additionally, the gene required for farnesol biosynthesis (*DPP3*) was upregulated 3-fold in comparison to control (Table 2). Primers used for real-time PCR (Table 3).

## 4. Discussion

The poor efficacy and severe side effects of common antifungal treatments and the occurrence of multidrug-resistant *Candida* strains require an alternative means to the regular use of antifungal agents. Since the fungal pathogenicity is mostly attributed to their capacity to adhere and to accumulate on various surfaces, the novel antifungal strategy should focus on the development of agents that prevent biofilm formation and also eradicate existing biofilms. The ability of *C. albicans* to form biofilm on mammalian cell surfaces as well as on implanted medical devices is an essential virulence determinant that stimulates progress of the infection.

Numerous plant-derived compounds have been demonstrated to exhibit potential anti-*Candida* activities via multiple mechanisms, including inhibition of yeast-to-hyphae transition; prevention of biofilm formation; impairing cell metabolism, cell wall integrity, and cell membrane fluidity; as well as initiation of apoptosis [35]. Our results clearly show strong dose- and time-dependent inhibition of *C. albicans* biofilm formation by the phytocannabinoid CBD. CBD had an MBIC_50_ at a dose of 12.5 μg/mL after 48 h and 72 h of incubation, and an MBIC_90_ of 100 μg/mL already at 24 h without any renewal of biofilm formation during the next 48 h. Cell membrane permeability assay performed on 24 h-old biofilms confirmed dose-dependent elevation of the damaged cells in biofilm upon CBD treatment. The CBD effect seems to be specific to the biofilms, since neither the viability nor the growth of planktonic fungi was significantly affected after 24 h even at 4 X MBIC_90_ [34]. Furthermore, the tested agent was capable to reduce notably biofilm thickness and EPS production, thus impairing biofilm architecture and integrity. The decrease in EPS production was linked to downregulation of *ADH5* gene responsible for production of extracellular matrix as well as *FKS1* and *BIG1* genes required for the synthesis of β-1,6-glucan—the main component of EPS.

Inhibition of biofilm formation by CBD was associated with modification of fungal morphology. *C. albicans* are classified as dimorphic fungi, which can transit between the two major forms—yeast and hyphae. The switch from yeast to hyphae is considered a critical virulence characteristic of *Candida*. The hyphae in biofilms support the development of stable structure. Therefore, preventing morphological transition or/and eliminating hyphal form of *C. albicans* can improve the antifungal therapy. Previously, we reported that the endocannabinoids AEA and AraS exhibited antifungal activity against *C. albicans* by inhibiting yeast-to-hyphae transition and hyphal growth [27]. In the present work, we observed that CBD caused a striking elevation of clustered yeast and appearance of pseudohyphae forms occurring in parallel with overall decrease in true hyphae mass. This was associated with a significant increase in yeast wall protein (*YWP1*) transcript along with moderate downregulation of hyphae-specific genes (*HWP1*, *ALS3*, *ECE1*) and genes essential for filamentation transcriptional regulators (*EFG1*, *RAS1*). Moreover, the expression of a key regulator of hyphal extension, *EED1*, was remarkably suppressed by CBD. This gene plays an important role in *C. albicans* host tissue invasion [36]. It should be emphasized that although pseudohyphae are physically more similar to hyphae, their properties are more relevant to yeasts, and therefore could be defined as elongated chains of yeast cells maintaining constrictions at the septal junctions [37,38]. *C. albicans* strains grown as pseudohyphae exhibited reduced virulence as compared to true hyphae [39]. We detected significant upregulation of heat shock protein *HSP90* gene expression that negatively regulates hyphal development through repressing Ras1-cAMP-PKA signaling pathway [40]. In addition, we observed pronounced removal of pre-formed biofilm treated with CBD. The biofilm eradication action of CBD is especially important, since *C. albicans* biofilms are often associated with indwelling medical devices, aggravating the severity of the disease and the mortality rate of affected patients [41].

Multiple virulence factors of *C. albicans* are associated with the infection-related host tissue damage processes such as extracellular hydrolytic enzymes and hyphae-specific cell surface proteins. In this study, we tested the expression of virulence-related genes such as lipases, phospholipases, and cell wall proteins related to the RBT family in *C. albicans* biofilms formed after a 24 h incubation with CBD. The extracellular lipases produced by *C. albicans*, LIPs, contribute to the delivery of nutrients and support fungal permeation of the host tissues [42]. Phospholipase B (PLBs) are also involved in the virulence of *C. albicans* by assisting the fungi in crossing host cell membranes with its following destruction [43]. RBTs genes are induced during filamentous growth and contribute to fungal pathogenesis [44]. In this study, we observed noticeable downregulation of all examined virulence genes (*LIP2*, *LIP4*, *LIP5*, *PLB1*, *PLB2*, *RBT1*, *RBT4*, *RBT5*) in biofilms formed in the presence of CBD.

Based on the above findings, we further studied the mode of anti-biofilm activity of CBD at sub-MBIC_90_. Our findings demonstrate significant alteration of mitochondrial activity upon fungal treatment with CBD. CBD induced intracellular ROS production with concomitant reduction in the antioxidant defense genes *SOD* (*SOD1*, *SOD2* and *SOD4*) and *TRR1*. This might explain the inability of *C. albicans* to tolerate CBD-induced oxidative stress. Additionally, the anti-biofilm agent Thiazolidinedione-8 impairs *C. albicans* protection mechanism against oxidative stress in *C. albicans*–*Streptococcus mutans* mixed biofilms [32]. In general, elevated intracellular ROS in fungi triggers mitochondria abnormal function [45,46] and decreases ATP generation [47]. Consistent with this, we observed a CBD-induced increase in MMP indicative for hyperpolarization of the mitochondrial membrane with concomitant decrease in ATP levels. Additionally, some other compounds have been shown to elevate MMP and decrease ATP production in yeast resulting in ROS accumulation which triggers mitochondrial dysfunction and apoptosis [48,49]. Another notable finding was related to the pronounced downregulation of the Ca^2+^/H^+^ antiporter *VCX1*, which is located on the fungal vacuole membrane and regulates the concentration of cytosolic Ca^2+^ by calcium uptake into the vacuole, while releasing H^+^ ions into the cytosol [50]. Since Vcx1 acts to lower Ca^2+^ in yeast [51], *VCX1* downregulation by CBD can cause elevation of cytosolic Ca^2+^ which subsequently impairs mitochondrial function. Numerous studies have demonstrated that oxidative stress induced by natural compounds leads to elevation of intracellular Ca^2+^, which in turn triggers mitochondrial dysfunction and subsequent apoptosis of *C. albicans* cells [52,53,54,55,56].

Due to the reduced expression of genes involved in cell wall integrity and maintenance (chitin synthase *CHS1* and chitinases *CHT1*, *CHT2*, and *CHT3*), we proposed that CBD at sub-MBIC_90_ causes cell wall stress. Indeed, in our study, the decrease in chitin production and the appearance of branched unseparated pseudohyphal cells was correlated with notable downregulation of genes responsible for chitin synthesis (*CHS1*) and cell separation (*CHT3*). According to this observation, we suggest that the altered biofilm morphology might be explained by CBD-induced cell wall stress manifested in reduction in chitin production and defective morphogenesis. As mentioned above, treatment of biofilm with CBD also leads to elevated levels of mRNA of *HSP* genes, which are overexpressed in adaptive response to cellular stress during biofilm formation [57]. Furthermore, correlation between upregulation of *HSP90* gene and cell death in fungal biofilm upon CBD exposure can be explained by tight relation of this gene to apoptosis [58]. In accordance with our findings, CBD was shown to induce death of acute lymphoblastic leukemia cells via a similar mode of action: targeting mitochondrial dysfunction, oxidative stress, and altering calcium homeostasis [59].

Ergosterol is an important virulence factor contributing to normal functioning and structural integrity of the plasma membrane in *Candida* species [60]. In our study, genes related to ergosterol biosynthesis, *ERG11* and *ERG20*, were downregulated by sub-MBIC_90_ of CBD which could be attributed to its cell membrane-targeting activity. On the other hand, we recorded notable upregulation of *DDP3* gene associated with biosynthesis of farnesol—the molecule able to inhibit *C. albicans* biofilm formation [61]. Similarly, other natural products have been demonstrated to inhibit *C. albicans* biofilm formation by upregulating *DPP3* gene and by downregulating ergosterol biosynthesis-associated genes [62].

In conclusion, we demonstrate here that CBD exhibits specific anti-biofilm activity against C. albicans without significantly affecting either planktonic fungal growth or viability. We propose that CBD exerts its anti-biofilm activity towards *C. albicans* biofilm through a multi-target mode of action, which differs from commonly used antimycotics. Thus, CBD should be explored for further development as an alternative treatment to combat fungal infections.

## Figures and Tables

**Figure 1 microorganisms-09-00441-f001:**
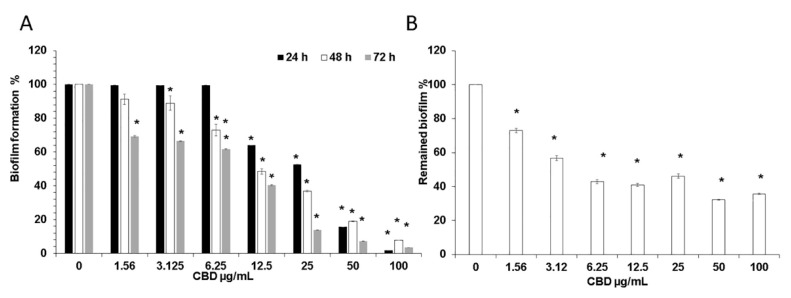
(**A**) Effect of CBD on *C. albicans* biofilm formation. *C. albicans* yeast cells were incubated with various concentrations of CBD for 24, 48, and 72 h at 37 °C; (**B**) Effect of CBD on preformed biofilms. Biofilms of *C. albicans* which were 24 h-old were exposed for 24 h to various concentrations of CBD and the metabolic activity of biofilm cells was measured using the MTT assay. The values of the untreated control were set to 100%. * Significantly lower than the untreated control (*p* < 0.05).

**Figure 2 microorganisms-09-00441-f002:**
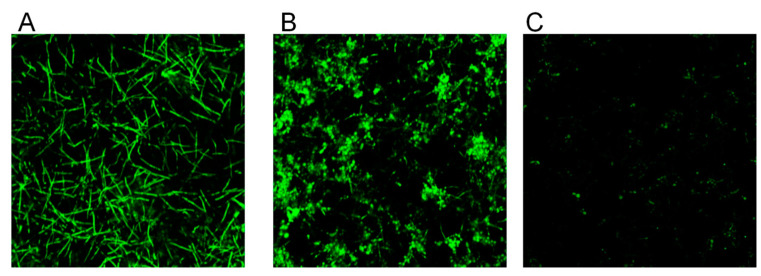
Effect of CBD on biofilm morphology. A–C. *C. albicans*–GFP biofilm were formed for 24 h in the absence of (**A**), presence of 25 μg/mL (**B**), or 100 μg/mL (**C**) CBD. The morphology of biofilms was visualized using NIKON confocal microscope with 488 nm laser excitation and collecting the data using the 535 nm green filter. Magnification 100×.

**Figure 3 microorganisms-09-00441-f003:**
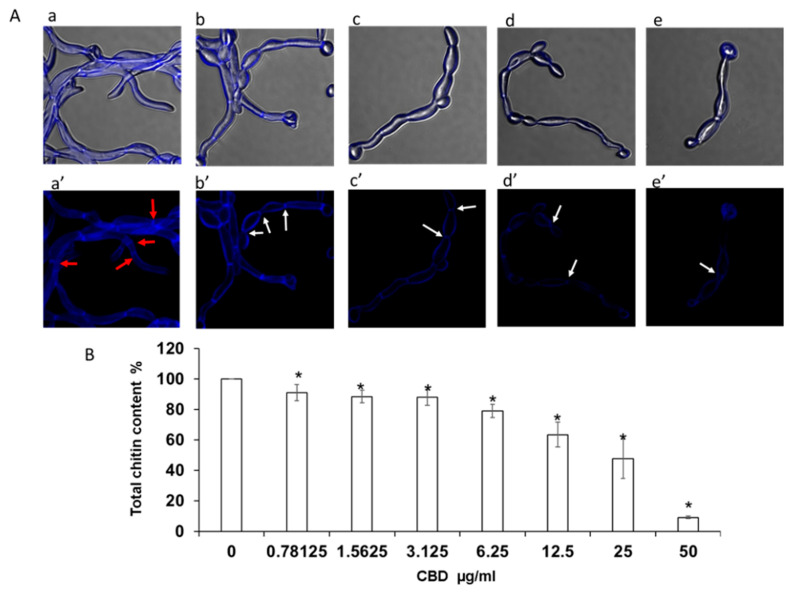
Determination of chitin content in biofilms. *C. albicans* biofilms after exposure to CBD for 24 h were stained with calcofluor white (CFW) for 5 min in the dark. (**A**). CFW-stained biofilms were observed by confocal laser scanning microscopy (CLSM). Phase-contrast and CFW stain merged images (a–e); CWF stain (a’–e’) of untreated control (a,a’); biofilms treated with 6.25 μg/mL (b,b’), 12.5 μg/mL (c,c’), 25 μg/mL (d,d’), 50 μg/mL (e,e’) of CBD. Red and white arrows indicate true septa and constrictions at the septal junctions, respectively. Magnification 600×. (**B**). Total chitin content was measured according to the CFW fluorescence intensity measured in a plate reader. Data are presented as percentage of the untreated control. * Significantly lower than the untreated control (*p* < 0.05).

**Figure 4 microorganisms-09-00441-f004:**
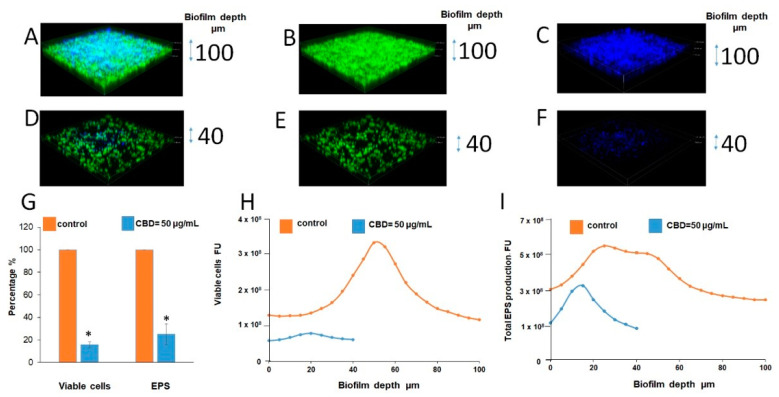
CLSM of biofilms. (**A**–**F**). CLSM 3D images of untreated and treated with CBD biofilms. (**A**,**D**). Merged images of metabolically active green fluorescence protein (GFP)-expressing (green color) fungal cells embedded in the extracellular polysaccharides (EPS) (blue color, ConA-stained); (**B**,**E**). Images of fungal cells alone (green color); **C**, **F**. Images of EPS alone (blue color); (**A**–**C**). Untreated control; (**D**–**F**). Biofilms were developed in the presence of 50 μg/mL CBD. Z-axis of 3D images indicates biofilm thickness. Magnification 100×. (**G**–**I**). Quantitative analysis of total EPS production and viable cells amount in each sample was calculated as blue and green fluorescence intensity, respectively, using Image J v3.91 software. Data are presented as total EPS production (**I**) and viable cells amount (**H**) in each layer of the biofilm (5 µm sections). Percentage of total EPS production and viable cells in biofilms treated with 50 μg/mL CBD is presented as area under the curve (AUC) and compared to untreated control (**G**). * Significantly lower than the untreated control (*p* < 0.05).

**Figure 5 microorganisms-09-00441-f005:**
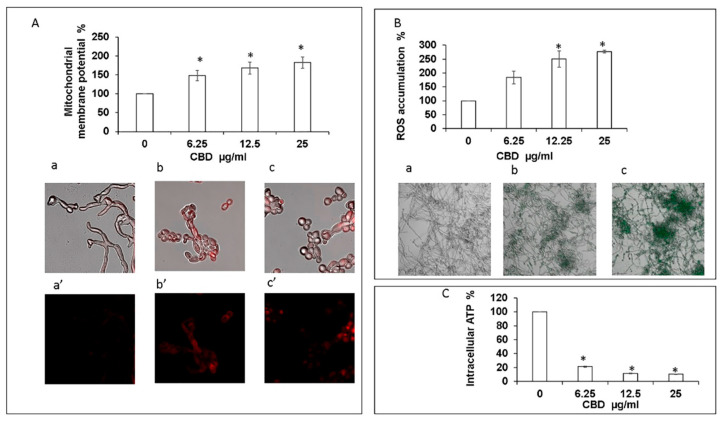
Assessment of mitochondrial function and ROS production. (**A**). Effect of CBD on *C. albicans* mitochondrial membrane potential (MMP) as determined by DiOC2(3). * Significantly higher than the untreated control (*p* < 0.05); Mito Tracker Red-stained biofilms were observed by CLSM. Phase-contrast and Mito Tracker Red stain merged images (a–c), or Mito Tracker Red stain (a’–c’) of untreated control (a,a’); biofilms treated with 12.5 μg/mL (b,b’) or 25 μg/mL (c,c’) of CBD. Magnification 600×. (**B**). Effect of CBD on ROS accumulation. *C. albicans* biofilms formed in the absence or presence of CBD were loaded with DCFHDA and the relative fluorescence intensities (RFI) of the biofilms were measured in a plate reader. The RFI values were normalized to the amount of metabolically active cells in biofilms assessed by MTT assay. * Significantly higher than the untreated control (*p* < 0.05). Fluorescence images of untreated control (a), biofilm treated with 6.25 μg/mL (b) and 25 μg/mL (c) of CBD were captured by CLSM. Magnification 200×. (**C**). Effect of CBD on intracellular ATP levels. *C. albicans* biofilms formed in the absence or presence of CBD were analyzed for ATP content using the BacTiter-Glo reagent. * Significantly lower than the untreated control (*p* < 0.05).

**Figure 6 microorganisms-09-00441-f006:**
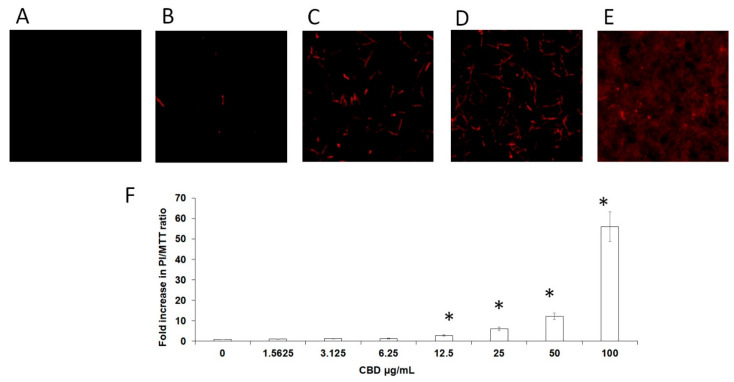
Membrane permeability of biofilm cells. *C. albicans* biofilms formed after a 24 h incubation with CBD were incubated with PI. A–E. PI-stained biofilms were observed by CLSM. PI staining of untreated control (**A**); biofilms treated with 3.12 μg/mL (**B**), 12.5 μg/mL (**C**), 50 μg/mL (**D**), or 100 μg/mL (**E**) CBD. Magnification 100×. (**F**). Relative PI fluorescence was measured in a plate reader. The ratios between dead cells and metabolically active cells in biofilms was calculated as PI fluorescence values divided on values obtained from MTT assay (PI/MTT ratio). Data are presented as a fold increase in PI/MTT ratio in treated biofilms compared to PI/MTT ratio in untreated biofilms. * Significantly lower than the untreated control (*p* < 0.05).

**Table 1 microorganisms-09-00441-t001:** Genes downregulated in CBD-treated *C. albicans* biofilm compared to their expression in untreated control as identified by real-time RT-PCR.

Gene	Fold Change	SEM ^a^	Description and Function ^b^
*adh5*	−2.69	0.04	Putative alcohol dehydrogenase; production of extracellular matrix
*big1*	−3.75	0.12	Endoplasmic reticulum protein; required for beta-1,6-glucan synthesis, filamentation, adhesion, and virulence
*FKS1*	−1.41	0.02	Essential beta-1,3-glucan synthase subunit; beta-1,3-glucan is the main component of extracellular matrix
*ece1*	−2.96	0.01	Candidalysin, cytolytic peptide toxin essential for mucosal infection; hypha-specific
*hwp1*	−1.46	0.05	Hyphal cell wall protein; hypha-specific
*eed1*	−6.21	0.28	RNA polymerase II regulator; filamentation
*efg1*	−1.64	0.02	bHLH transcription factor; hyphal growth, cell-wall gene regulation; roles in adhesion, virulence
*ras1*	−1.66	0.07	Regulates cAMP and MAP kinase pathways; role in hyphal induction, virulence, apoptosis
*ALS3*	−1.72	0.06	Cell wall adhesin; hyphal-specific
*chs1*	−2.54	0.12	Chitin synthase; essential; for primary septum synthesis in yeast and hyphae
*cht1*	−3.73	0.15	Chitinase; putative N-terminal catalytic domain; has secretory signal sequence
*cht2*	−1.75	0.01	GPI-linked chitinase; required for normal filamentous growth
*cht3*	−7.99	0.23	Major chitinase; cell separation
*erg11*	−3.37	0.23	Lanosterol 14-alpha-demethylase; ergosterol biosynthesis
*erg20*	−3.32	0.10	Putative farnesyl pyrophosphate synthetase involved in isoprenoid and sterol biosynthesis
*lip2*	−2.82	0.05	Secreted lipase; hydrolysis and synthesis of triacylglycerols
*lip4*	−1.59	0.04	Secreted lipase
*lip5*	−1.55	0.10	Cold-activated secreted lipase
*plb1*	−1.46	0.22	Phospholipase B; host cell penetration and virulence in mouse systemic infection
*plb2*	−3.03	0.06	Putative phospholipase B; tissue degradation, hyphal formation, and host invasion, hydrolytic activity
*rbt1*	−2.90	0.17	Cell wall protein with similarity to Hwp1; hypha-specific
*rbt4*	−3.93	0.30	Plant pathogenesis-related protein, may damage either host cells or competing microbes; hypha-specific
*rbt5*	−3.04	0.01	GPI-linked cell wall protein; hemoglobin utilization; required for RPMI biofilms
*pra1*	−9.47	0.70	Cell surface protein that sequesters zinc from host tissue; enriched at hyphal tips; released extracellularly
*sod1*	−3.44	0.04	Cytosolic copper- and zinc-containing superoxide dismutase; antioxidant defense
*sod2*	−2.55	0.10	Mitochondrial Mn-containing superoxide dismutase; protection against oxidative stress
*sod4*	−2.39	0.05	Cu-containing superoxide dismutase; antioxidant defense
*trr1*	−7.98	0.15	Thioredoxin reductase; antioxidant defense
*vcx1*	−2.80	0.11	Putative H^+^/Ca^2+^ antiporter; tolerance and virulence through calcineurinand Ca^2+^/H^+^ exchanger

^a^ Standard errors of the means, ^b^ as described in the CGD database (http://www.candidagenome.org/; access on 18 February 2021).

**Table 2 microorganisms-09-00441-t002:** Genes upregulated in CBD-treated *C. albicans* biofilm compared to their expression in untreated control as identified by real-time RT-PCR.

Gene	Fold Change	SEM ^a^	Description and Function ^b^
*HSP104*	2.29	0.02	Heat-shock protein; heat shock/stress induced
*HSP70*	2.25	0.03	Heat-shock protein; heat shock/stress induced, response to toxic substance
*HSP90*	2.65	0.05	Heat-shock protein; cellular response to drug, negative regulation of filamentous growth, key regulator of biofilm dispersion
*YWP1*	40.25	0.11	Secreted yeast wall protein; antiadhesive effect, yeast-specific
*DPP3*	3.05	0.15	Required for farnesol biosynthesis

^a^ Standard errors of the means, ^b^ as described in the CGD database (http://www.candidagenome.org/; 18 February 2021).

**Table 3 microorganisms-09-00441-t003:** Primers used for real-time PCR.

Gene	Forward Primer	Reverse Primer
*act1*	AAGAATTGATTTGGCTGGTAGAGA	TGGCAGAAGATTGAGAAGAAGTTT
*adh5*	ACCTGCAAGGGCTCATTCTG	CGGCTCTCAACTTCTCCATA
*als3*	TAATGCTGCTACGTATAATT	CCTGAAATTGACATGTAGCA
*big1*	TTATTCGTCCTACTAGCAT	CATATTTGTCACCGAAGTAA
*chs1*	CTGACAAGAGCCAACACTGC	CGCCTCTTGATGGTGATGAT
*cht1*	CCTGTTGCTGCTACTACTAC	TTGTAGCATTTGGCTGCCCA
*cht2*	GCACCAAATACGTCACCATTG	GAAGGCAAAGGCAGCCAATAA
*cht3*	GTATTTCCAAATCCAGTTC	GTCAATATTTGATAAGTCG
*dpp3*	TTATCTGTAATTATCATTGT	GTTGTCAAACTTCAATTGA
*ece1*	GCTGGTATCATTGCTGATAT	TTCGATGGATTGTTGAACAC
*eed1*	AGCAACGACTTCCAAAAGGA	CGGTTTCTGGTTCGATGATT
*efg1*	TATGCCCCAGCAAACAACTG	TTGTTGTCCTGCTGTCTGTC
*erg11*	AAGAATCCCTGAAACCAA	CAGCAGCAGTATCCCATC
*erg20*	TTACCCGTGGCATTAGCAATGTA	TCCCAAGGGAATCAAAATGTCTC
*fks1*	CGTGAAATTGATCATGCCTGTAC	AACCCTTCTGGGCTCCAAA
*hsp70*	TGCCGTTGTTACCGTTCCAGCTTA	AACCATAAGCAATGGCAGCAGCAG
*hsp90*	GCTTTAAGTGCTGGTGCTGACGTT	TGGTACCACGACCCAATCTTTCGT
*hsp104*	ACGGCCATACTCTGTGGTCTTGTT	CGGCATTGATGTAGTTTGCACCCA
*hwp1*	CACAGGTAGACGGTCAAGGT	AAGGTTCTTCCTGCTGTTGT
*lip2*	GGCCTGGATTGATGCAAGAT	TTGTGTGCAGACATCCTTGGA
*lip4*	GCGCTCCTGTTGCTTTGACT	ACACGGTTTGTTTTCCATTGAA
*lip5*	TGGTTCCAAAAATACCCGTGTT	CGACAATAGGGACGATTTGATCA
*plb1*	GGTGGAGAAGATGGCCAAAA	AGCACTTACGTTACGATGCAACA
*plb2*	TGAACCTTTGGGCGACAACT	GCCGCGCTCGTTGTTAA
*pra1*	GCTTTGGATGTGTATGCATATG	CTAGGGTTGCTATCGGTATGTG
*ras1*	GGCCATGAGAGAACAATATA	GTCTTTCCATTTCTAAATCAC
*rbt1*	CTGCAAAAACAGTGCTCTCG	CAAGAATGCAGCAAGACCAA
*rbt4*	ATCGCCTATGTCACCCAGAC	CATTACCACCATCAGCATCG
*rbt5*	CTGCTAAAGAAACCACTGCTG	GCTTCAACGGAAACAGAAGC
*sod1*	TTGAACAAGAATCCGAATCC	AGCCAATGACACCACAAGCAG
*sod2*	ACCACCCGTGCTACTTTGAAC	GCCCATCCAGAACCTTGAAT
*sod4*	CCAGTGAATCATTTGAAGTTG	AGAAGCACTAGTTGATGAACC
*trr1*	TTTCTGCCTGTGCTGTTTGT	TTCCTGGAGTGGTTTGAATGTA
*vcx1*	CTTTGTCGCTGGTGGGATT	GCATCGGCACTTTGGTCT
*ywp1*	GCTACTGCTACTGGTGCTA	AACGGTGGTTTCTTGAC

## Data Availability

Raw data are available upon request.

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
