# Peer review of "Anti-Biofilm Activity of Cannabidiol against Candida albicans"

_microorganisms, 2021, doi:10.3390/microorganisms9020441_

Round 1

Reviewer 1 Report

The manuscript presented is thorough and methodically of high standard. The results presented are clear and statistically representative. The discussion is comprehensive based on the data presented.

I have two minor comments:

  • The final % of ethanol in each of the assay presented is not included in methods. It is presumably constant through out the various concentrations tested as well as in the untreated controls: is it constant and what is it?
  • Lines 313 t 320 are in the present tense while the rest of the manuscript is in the past tense.

Author Response

Reviewer

The final % of ethanol in each of the assay presented is not included in methods. It is presumably constant through out the various concentrations tested as well as in the untreated controls: is it constant and what is it?

Author's Notes

The highest final concentration of ethanol in all assays was 1%. In order to determine the effect of ethanol on C. albicans, samples containing % of ethanol equal to the highest % of ethanol in CBD-treated sample were compared in every experiment to controls without ethanol. No significant difference between ethanol-treated and untreated C. albicans was observed.

Reviewer

Lines 313 t 320 are in the present tense while the rest of the manuscript is in the past tense.

 Author's Notes

Corrected in the text (in yellow)

Reviewer 2 Report

The manuscript by Feldman et al described the antibiofilm activity of cannabidiol against Candida albicans and study its mode of action. Globally, the study has been well conducted but the following points should be cleared before acceptance of this paper. 

My global remark lies on the absence of any reference in the assays. Indeed, why the authors did not have used any knwon antibiofilm agent in their assay? It would have helped to decipher if CBD is a great antibiofilm agent or a moderate one. 

L60-61 : The authors mentioned that cannabinoids have already been described as antimicrobial compounds. In parallel, they mentioned that, in their hands, no activity against planktonic C. albicans has been obtained (data not shown). I suggest that the authors add their data and also give the reported values of MIC obtained on relevant pathogens. It will be clearest to compare the antimicrobial activity. 

L64 : they should give the structure of cannabigerol

L93 : The authors should mention the number of replicates they did for the M&M part.

L270-272 : The paragraph is not clear. Do the authors refer to antimicrobial activity towards planktonic C. albicans or something else ? 

In conclusion : the authors propose a multi-target mode of action. COuld they envision any toxicity or resistance linked with this proposed mode of action? 

Moreover, from these data, do the authors plan activity of CBD against the emerging superbug C. auris ? 

In addition, some typos have to be corrected :

L70, 74 : additionnal spaces have to be removed

L471: via in italic

Author Response

Reviewer

My global remark lies on the absence of any reference in the assays. Indeed, why the authors did not have used any knwon antibiofilm agent in their assay? It would have helped to decipher if CBD is a great antibiofilm agent or a moderate one. 

Author's Notes

The focus of the present study was to understand the antibiofilm action mechanism of the specific compound CBD against C.albicans. Therefore, a comparative study regarding other known antibiofilm agents was not included. However, it would be of great interest to compare antibiofilm activity between known antibiofilm agents and CBD. Furthermore, the combinatory studies between CBD and these agents will reflect potential combinations of the agents against microbial biofilms. Further research is required to investigate the above issue.

Reviewer

L60-61 : The authors mentioned that cannabinoids have already been described as antimicrobial compounds. In parallel, they mentioned that, in their hands, no activity against planktonic C. albicans has been obtained (data not shown). I suggest that the authors add their data and also give the reported values of MIC obtained on relevant pathogens. It will be clearest to compare the antimicrobial activity. 

Author's Notes

We have added text (in yellow) to the Introduction section to explain this issue in more detail.

Reviewer

L64 : they should give the structure of cannabigerol

Author's Notes

We have added the CBG structure in the Introduction section

Reviewer

L93 : The authors should mention the number of replicates they did for the M&M part.

 Author's Notes

This has been added to the M/M section (in yellow)

Reviewer

L270-272 : The paragraph is not clear. Do the authors refer to antimicrobial activity towards planktonic C. albicans or something else ?

 Author's Notes 

Yes, we evaluated the effect of CBD towards planktonic fungi (added in the text in yellow)

Reviewer

In conclusion : the authors propose a multi-target mode of action. COuld they envision any toxicity or resistance linked with this proposed mode of action? 

 Author's Notes

CBD is a FDA-approved drug and CBD-based formulation has been proven safe for human administration. Blaskovich and colleagues demonstrated that CBD does not lead to bacterial resistance after repeated exposure (DOI:https://doi.org/10.1038/s42003-020-01530-y). CBD is a non-psychotic agent of the cannabinoid family and is sold also OTC in countries in Europe and some states in USA.

We observed that CBD, in contrast to common antifungal agents, does not affect significantly either the growth or the viability of C. albicans. Rather, it targets various fungal properties essential for virulence. Therefore, it seems that CBD does not cause fungal resistance, however, further research is requested to prove it.

Reviewer

Moreover, from these data, do the authors plan activity of CBD against the emerging superbug C. auris ? 

 Author's Notes

Candida auris is an emerging type of fungus that is a world-wide health threat as it is a potential multidrug-resistant fungus. It will be indeed very interesting to check if the  effect of CBD is also on resistant fungi. 

Reviewer

In addition, some typos have to be corrected :

L70, 74 : additionnal spaces have to be removed

 Author's Notes

done

Reviewer

L471: via in italic

 Author's Notes

done

Round 2

Reviewer 2 Report

I have no particular comment. The authors answered the points I suggested in my review.